# From Player to Master: Enhancing Test-Time Learning of LLM Agents via Reinforcement Learning over Memory

**Yishuo Cai** [1]  **Xingyu Guo** [2]  **Xuancheng Huang** [3]  **Jinhua Du** [4]  **Can Huang** [3]  **Wenxuan Huang** [5]  **Wenhan Ma** [1]
**Yuyang Hu** [6]  **Aohan Zeng** [4]  **Jie Tang** [4]  **Xu Sun** [1]

## Abstract

Large language model (LLM) agents are increasingly deployed in long-running settings where improving through experience at test time becomes important. A common approach is to update an explicit memory after each interaction to guide future decisions. However, most existing methods rely on hand-designed prompting rules, making it difficult to align memory updates with downstream objectives over multistep horizons consistently. We propose MEMO-PILOT, a plug-in memory copilot that *explicitly trains* the memory update process to improve a frozen LLM's performance across sequential interactions. We formulate memory updating as a multi-turn decision problem and optimize it end-to-end with multi-turn GRPO. Our training recipe introduces (i) a turn-wise reward signal and (ii) a context-independent, turn-level advantage estimation across rollouts, enabling finer-grained credit assignment and more stable training in multi-turn settings. We evaluate MEMOPILOT on two testbeds: multi-round Rock–Paper–Scissors (RPS) and Limit Texas Hold'em (LHE). Across both environments, MEMOPILOT substantially improves test-time learning of a frozen player over strong baselines, ranking first in Elo ratings on both games (1762 on LHE and 1590 on RPS) and outperforming all baseline memory methods and proprietary models, including DeepSeek-V3.2. Our code is publicly available here.

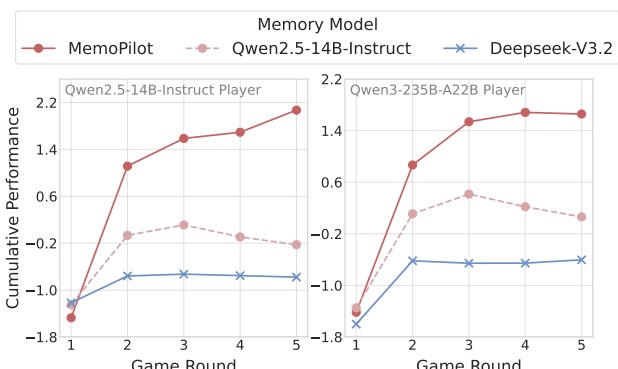

*Figure 1.* Test-time learning dynamics in Limit Texas Hold'em (LHE) of our memory model (MEMOPILOT) compared to baseline memory models across sequential games, evaluated with two frozen players. Cumulative performance denotes the running average of per-game scores up to the current round. Left: the player used during memory training (Qwen2.5-14B-Instruct). Right: zero-shot generalization to a stronger frozen player (Qwen3-235B-A22B). MEMOPILOT yields consistently higher cumulative performance and improves rapidly within a few games.

## 1. Introduction

Large language model (LLM) agents are increasingly used in settings that involve repeated interactions with related tasks, users, or environments. In such settings, a key capability is *test-time learning* (TTL), where an agent improves over a sequence of interactions by leveraging experience accumulated during deployment. Recent benchmarks and analyses have begun to systematically evaluate such learning capability and efficiency in LLMs and agents (Dou et al., 2025; Zheng et al., 2025b; Wang et al., 2025a), highlighting that the ability to leverage experience can be a central bottleneck for real-world agent reliability and efficiency. This motivates memory-aware agent systems that can accumulate and exploit experience online to improve future decisions.

A growing line of work attempts to realize TTL via explicit memory and experience-driven adaptation. Early approaches such as Reflexion (Shinn et al., 2024) and ExpeL (Zhao et al., 2024) demonstrate that agents can iteratively improve by reflecting on interactions and accumulating experience. More recent methods move beyond static

[1]Peking University, Beijing, China [2]Central South University, Changsha, China [3]Zhipu AI, Beijing, China [4]Tsinghua University, Beijing, China [5]East China Normal University, Shanghai, China [6]Renmin University of China, Beijing, China. Correspondence to: Xuancheng Huang <xchuang17@163.com>, Xu Sun <xusun@pku.edu.cn>.

*Proceedings of the 43rd International Conference on Machine Learning*, Seoul, South Korea. PMLR 306, 2026. Copyright 2026 by the author(s).

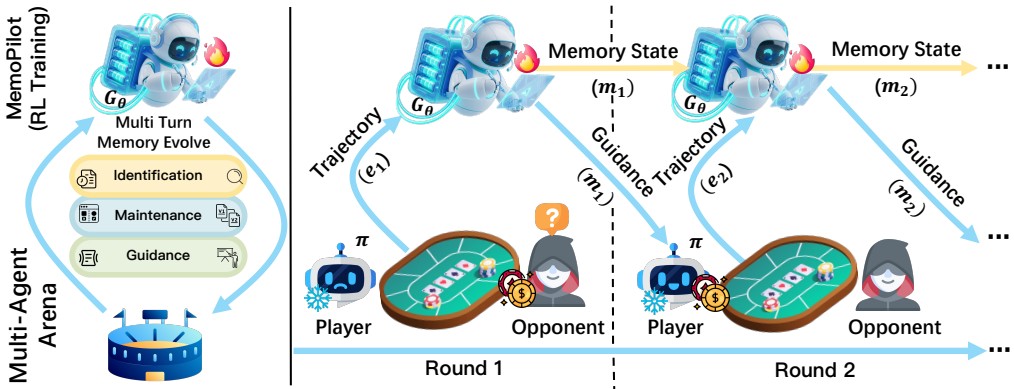

*Figure 2.* An overview of the MEMOPILOT framework. A trainable memory model $G_\theta$ iteratively updates a memory state $m_t$ from interaction trajectories and provides it as guidance to a plug-and-play frozen player $\pi$. All cross-game learning depends on the evolving memory, while $\pi$ remains unchanged.

storage or naive history reuse and start to incorporate *dynamic* updates: Dynamic Cheatsheet (Suzgun et al., 2025) maintains an evolving memory for test-time adaptation; ReasoningBank (Ouyang et al., 2026) distills reusable reasoning strategies from an agent's successes and failures and closes the loop via retrieval and consolidation. Together, these works suggest that *dynamic memory update* is a promising interface for enabling TTL.

However, despite these advances, most existing approaches rely on hand-designed or prompt-based memory update rules, rather than end-to-end optimization of the memory update policy (Suzgun et al., 2025; Ouyang et al., 2026). In our pilot observations, even strong instruction-following LLMs fail to consistently improve across repeated interactions when memory updates are driven only by such heuristic mechanisms, motivating a training signal that directly optimizes memory updates for downstream performance. More broadly, learning to improve at test time has rarely been treated as a trainable capability.

To address this gap, we propose MEMOPILOT, a plug-in **Memo**ry Co**pilot** that explicitly trains the memory update process to improve the performance of a frozen LLM in multi-turn interactions. Inspired by Suzgun et al. (2025), we view memory as an evolving artifact that refines across multiple interactions. We treat memory updating as a trainable multi-turn decision problem and optimize it end-to-end with multi-turn GRPO (Shao et al., 2024). Concretely, we introduce a *turn-wise* reward signal and a *turn-level* advantage estimation across rollouts, which provides finer-grained credit assignment and stabilizes learning in multi-turn settings. This approach yields a natural *proxy task* where memory quality is assessed by downstream task performance. Multi-turn training is essential as it teaches memory up-

date through an iterative "**hypothesize-and-verify**" cycle: observes evidence from the current experience, proposes or refines hypotheses, verifies them against accumulated evidence, and corrects prior conclusions.

We evaluate MEMOPILOT on two strategic games including multi-round Rock–Paper–Scissors (RPS) (Guertler et al., 2025) and Limit Texas Hold'em (LHE) (Zha et al., 2019) because they closely match the TTL setting and satisfy three desiderata: (i) *learnability under cross-game interaction*: there exists exploitable, opponent-specific behavioral structure that can be discovered from multi-game experience; (ii) *controllability*: opponents can be specified by explicit strategies, enabling reproducible interactions and systematic coverage/generalization tests; and (iii) *challenge with measurable reward*: both environments provide clear outcome rewards suitable for end-to-end optimization, yet require non-trivial adaptation. LHE introduces imperfect information and rich hand-level variation that acts as natural probes of opponent behavior. While RPS has a small action space, multi-round interaction induces history-dependent dynamics; by designing diverse rule-based and mixed-strategy opponents, it remains challenging while allowing scalable and controlled construction of opponent families. Across both testbeds, we show that plugging MEMOPILOT into a frozen player substantially improves test-time learning performance over strong baselines.

Our main contributions are: (1) We propose MEMOPILOT, a plug-in memory pilot that improves a frozen LLM player's test-time learning behavior across repeated interactions by training the memory update process end-to-end. (2) We introduce a multi-turn GRPO training recipe for memory updating with turn-wise rewards and turn-level advantage estimation, enabling stable credit assignment in multi-turn

test-time learning rollouts. (3) We validate MEMOPILOT on controlled game testbeds, demonstrating consistent gains in test-time learning.

## 2. Preliminaries

Test-time learning (TTL) studies settings where an agent receives a stream of related tasks or interactions and improves its performance over time by leveraging experience accumulated during deployment. The stream is revealed sequentially (without access to future interactions), so adaptation must be done online based on past experience.

In this work, we focus on a sequential-game TTL setting for strategic interactions. Here, each TTL unit is a game (or match) played against an opponent, and the agent is evaluated by the game outcome (e.g., win/loss or chip gain), providing a natural reward signal. Crucially, opponents exhibit exploitable strategy structure, making cross-game adaptation meaningful: information inferred from earlier games can improve decisions in later games.

**Notation.** We denote the sequence of games by $\{g_t\}_{t=1}^T$. Game $g_t$ yields an interaction trajectory $e_t$ and a scalar reward $r_t \in \mathbb{R}$. We consider a memory-based TTL formulation where learning depends on an explicit textual memory updated online without updating model parameters (Suzgun et al., 2025; Ouyang et al., 2026). A memory model $G_\theta$ reads accumulated experience and produces a textual memory

$$m_t = G_\theta(e_t, m_{t-1}), \quad m_0 = \emptyset. \quad (1)$$

which is provided to a fixed player model $\pi$ in subsequent games. The player itself is *stateless* across games: in each game, it only conditions on the current memory. The interaction evolves as

$$e_1 \sim \pi(\cdot \mid m_0), \quad m_1 = G_\theta(e_1, m_0),$$
$$e_{t+1} \sim \pi(\cdot \mid m_t), \quad m_{t+1} = G_\theta(e_{t+1}, m_t).$$

## 3. Method

We now present MEMOPILOT, a dynamic experiential memory model trained via multi-turn reinforcement learning. Given the sequential-game TTL setup in Sec. 2, we view memory updating as a sequential decision process, where the generator must learn to extract and express strategic insights that maximize the agent's cumulative performance across an episode of games.

### 3.1. Multi-Turn Memory Generation as a Markov Decision Process (MDP)

Following Sec. 2 and Eq. 1, we cast multi-turn memory updating as a sequential decision problem $\mathcal{M} = (\mathcal{S}, \mathcal{A}, \mathcal{P}, \mathcal{R})$,

where the memory model acts as a policy that must balance information extraction and strategic guidance across multiple interactions (Wang et al., 2025b).

Formally, the **state space** $\mathcal{S}$ consists of observation tuples $s_t = (e_t, m_{t-1})$, where $e_t$ is the latest game trajectory and $m_{t-1}$ is the previous memory. The **action space** $\mathcal{A}$ is the space of textual memories, and the generator samples $m_t \sim G_\theta(\cdot \mid s_t)$. We associate each game with an environment instance $E_t$ (e.g., poker private/public cards and positions), sampled from an environment distribution and varying across turns, capturing partial observability. The **transition dynamics** $\mathcal{P}$ is induced by the frozen player $\pi$ interacting with the opponent under $E_{t+1}$ conditioned on $m_t$, yielding the next trajectory $e_{t+1}$ and scalar reward $r_{t+1}$. The **reward function** $\mathcal{R}$ returns the observed game outcome $r_t$ for turn $t$ (a task-defined scalar).

An episode unfolds as $T$ games with interleaved memory updates following Eq. 1. The player then uses $m_t$ in game $t+1$. Since the first game serves as initial exploration without learned guidance, we define the episode return as the sum of rewards from the memory-guided games:

$$R(\tau) = \sum_{t=1}^{T-1} r_{t+1}, \quad (2)$$

where $\tau$ denotes the full trajectory. The training objective is to maximize expected return over opponent strategies $\sigma$ and trajectories $\tau$ generated by the memory policy:

$$\theta^* = \arg\max_\theta \ \mathbb{E}_{\sigma,\tau}\left[R(\tau)\right]. \quad (3)$$

To make optimization practical in multi-turn, stochastic environments, we use a turn-level, low-variance one-step proxy signal for advantage estimation that attributes outcomes to the most recent memory update, improving training stability and sample efficiency.

### 3.2. Training with Multi-Turn GRPO

To optimize the objective in Eq. 3, we adopt Group Relative Policy Optimization (GRPO) (Shao et al., 2024), which has proven effective for training LLM agents in multi-turn settings (Yu et al., 2026a). In the rollout phase, the policy model $G_{\theta_{\text{old}}}$ generates $G$ parallel episode rollouts for each opponent strategy $\sigma$. Each episode $i$ produces $T-1$ memory generations $\{m_{i,1}, m_{i,2}, \ldots, m_{i,T-1}\}$, where each $m_{i,t}$ decomposes into tokens $(m_{i,t,1}, m_{i,t,2}, \ldots, m_{i,t,|m_{i,t}|})$. Let $\{R_{i,t}\}_{i=1}^G$ denote the per-turn rewards at turn $t$. The group-normalized advantage is:

$$\hat{A}_{i,t,k} = R_{i,t} - \text{mean}(\{R_{i,t}\}_{i=1}^G), \quad R_{i,t} = r_{i,t+1}. \quad (4)$$

Following Liu et al. (2025b), we omit standard deviation normalization. This turn-specific advantage is applied to all tokens within the same memory generation step.

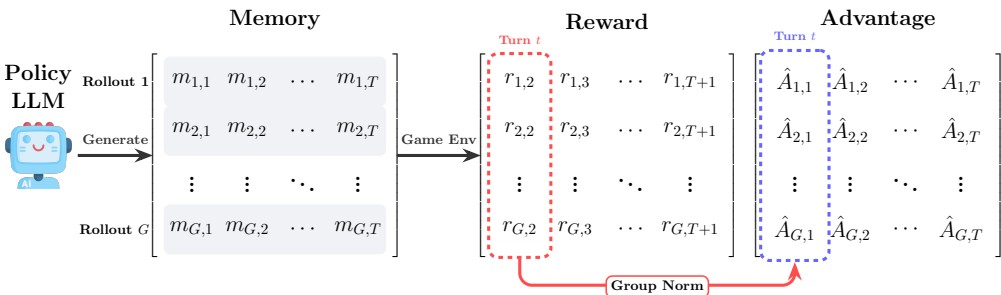

*Figure 3.* Multi-turn GRPO for memory updating with one-step (next-game) proxy rewards and **turn-level** advantages. Each rollout $i$ produces a sequence of memory states $\{m_{i,t}\}$; we assign each turn $t$ a one-step proxy return $R_{i,t} = r_{i,t+1}$ and compute a turn-level group-normalized advantage $\hat{A}_{i,t} = R_{i,t} - \text{mean}(\{R_{i,t}\}_{i=1}^{G})$, which is applied to the tokens of $m_{i,t}$.

While Eq. 3 optimizes the cumulative episode return, in practice we estimate turn-level advantages using the one-step outcome $R_{i,t} = r_{i,t+1}$. Using long-horizon returns would couple the learning signal to future stochasticity (e.g., different dealt hands), amplifying environment noise and making credit assignment unstable. The one-step proxy avoids this issue and yields a cleaner turn-wise learning signal, improving stability and sample efficiency for context learning.

As our approach spans multiple turns, each episode generates $T - 1$ context-independent memory updates. Inspired by Yu et al. (2026a), we optimize each memory generation step, extending the loss from the standard (group, token) structure to (group, turn, token). Let $r_{i,t,k}(\theta)$ denote the importance sampling weight for the $k$-th token of memory $m_{i,t}$:

$$r_{i,t,k}(\theta) = \frac{G_\theta(m_{i,t,k} \mid e_{i,t}, m_{i,t-1}, m_{i,t,<k})}{G_{\theta_{\text{old}}}(m_{i,t,k} \mid e_{i,t}, m_{i,t-1}, m_{i,t,<k})}. \quad (5)$$

The multi-turn GRPO objective with clipped surrogate and token-level averaging is:

$$\mathcal{J}(\theta) = \mathbb{E}_{\sigma \sim \mathcal{S}, \{m_{i,t}\} \sim G_{\theta_{\text{old}}}} \left[ \frac{1}{\sum_{i=1}^{G} \sum_{t=1}^{T-1} |m_{i,t}|} \right.$$
$$\left. \sum_{i=1}^{G} \sum_{t=1}^{T-1} \sum_{k=1}^{|m_{i,t}|} \mathcal{C}_{i,t,k} \right],$$
$$\text{where} \quad \mathcal{C}_{i,t,k} = \min \Big( r_{i,t,k}(\theta) \hat{A}_{i,t,k}, $$
$$\text{clip}\big(r_{i,t,k}(\theta), 1 - \varepsilon, 1 + \varepsilon\big) \hat{A}_{i,t,k} \Big).$$
$$(6)$$

The complete training procedure is summarized in Algorithm 1 (see Appendix A).

**Defining the Memory Space.** To support iterative refinement, we structure the memory space into three components:

(1) a diagnostic analysis that summarizes the evidence from recent interactions and updates hypotheses about the opponent strategy (*Identification*); (2) an explicit maintained belief state that records the current hypotheses and their confidence or verification status across turns under a fixed memory budget (*Maintenance*); and (3) concise, actionable guidance that the frozen player can execute in the next game (*Guidance*). During inference, these components enable an iterative update process: the generator revises its diagnosis and maintained beliefs as new evidence arrives, and updates the guidance accordingly. In addition, the verification or confidence signal in the maintained state provides a natural stopping criterion: once the hypothesis is sufficiently confirmed, the agent can continue playing without further memory revision. See Appendix D.1 for the exact prompt template and Appendix F for a multi-turn qualitative example of how the memory evolves.

### 3.3. Opponent Construction

A key design choice in our framework is constructing a diverse yet controllable opponent pool that enables systematic study of test-time learning. We design the opponent pool under three principles. *Controllability*: we specify each opponent using executable instructions to enable reproducible rollouts for stable RL training and evaluation. *Behavioral diversity*: for LHE, we vary action-frequency biases, street-specific aggression profiles, and deceptive modes (e.g., check-raise traps), while for RPS we cover open-loop sequences, one-step reactive rules, and multi-step counter-patterns. *Mechanism-based train–test separation*: held-out strategies preserve strategic intent while shifting triggers, or the phase where information is revealed, which probes whether memory can maintain and revise hypotheses as evidence accumulates.

Our construction follows a human-in-the-loop pipeline: experienced players write seed strategies, LLM-based rewriting expands and standardizes the set, and manual verification ensures each strategy is coherent and behaviorally

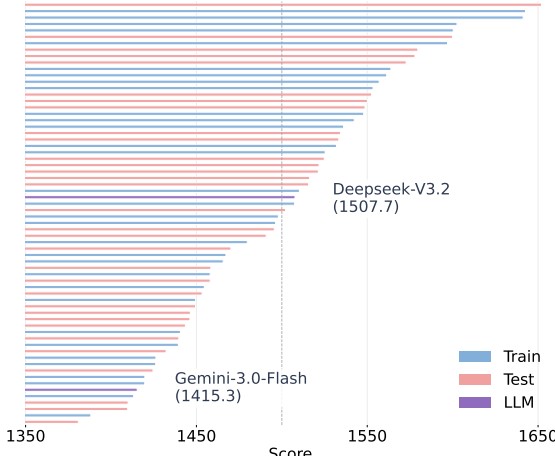

*Figure 4.* Elo ratings of RPS opponent strategies estimated from head-to-head matches, illustrating that our constructed opponent pool spans a broad and relatively uniform difficulty range. Blue and pink bars denote training and held-out opponents, respectively, while purple bars denote LLMs, while purple bars denote LLMs.

stable under our execution settings. Details of opponent construction and verification are provided in Appendix C.

**Elo-Based Difficulty Calibration.** We estimate an Elo rating for each opponent strategy via round-robin head-to-head matches to check that train/test pools span a broad difficulty range. Figure 4 shows the resulting Elo distribution for RPS, where train and held-out opponents cover a broad and relatively uniform difficulty range. We additionally include Gemini-3.0-Flash and DeepSeek-V3.2 (DeepSeek-AI et al., 2025) as reference baselines without access to specific strategies. We provide the corresponding Elo rating distribution for LHE in Appendix B.3 (Figure 8). We provide more implementation details in Appendix B.3.

## 4. Experiments

### 4.1. Experimental Setup

**Environments.** We evaluate two strategic games from TextArena (Guertler et al., 2025) and RLCard (Zha et al., 2019). The first is multi-round Rock–Paper–Scissors (RPS): each *game* contains 6 consecutive rounds, and both players observe the full history of previous rounds before making each decision. The second is Limit Texas Hold'em (LHE): each player has two private cards and chooses from four actions (Fold, Check, Call, Raise), featuring partial observability and stochastic outcomes.

**Metrics.** For RPS, we define the per-game *score* as the difference between the number of rounds won by the player and by the opponent over a 6-round match. We report RPS@k as the average per-game score over $k$ consecutive games. For LHE, a game consists of a duplicate match, in which

the players swap positions while sharing the same card deal. This duplicate-match eliminates variance induced by the card and seating order, enabling a fair comparison of players' strategic strength. We define the per-game chip as the sum of the final chip counts from these two subgames. We report LHE@k as the average per-game chip over $k$ consecutive games.

**Evaluation.** Due to the stochasticity of LLM sampling and game dynamics, we report results as mean@64 for both environments, averaging over 64 evaluation runs across strategies. All memory-based methods are evaluated with a fixed memory budget of 512 tokens. For LHE, we evaluate all methods on the same fixed set of card deals shared across evaluation runs to ensure a fair comparison.

**Training Details.** We use Qwen2.5-14B-Instruct as the base model of MEMOPILOT. We train a separate memory model for RPS and LHE. During training, we fix Qwen2.5-14B-Instruct as the player model. We also use it as the opponent model in both training and evaluation. Different opponents are constructed by providing different strategy system prompts. For evaluation, we assess the trained memory model's performance by pairing it with different player models. By default, one training rollout contains 3 consecutive games, during which the agent updates cross-game memory between games. For LHE, we use the same seed within each GRPO group so that rollouts share the same cards at the same turn. Computational cost is discussed in Appendix B.1.

**Baselines.** We compare MEMOPILOT against a set of baselines, including No Memory, Full History, Human-Written Counter-Strategy, Reflexion (Shinn et al., 2024), ExpeL (Zhao et al., 2024), MemoryBank (Zhong et al., 2023), AWM (Wang et al., 2025c), and ReasoningBank (Ouyang et al., 2026). No Memory plays $k$ independent games per opponent without any cross-game state. Full History provides the full interaction history from previous games as context. Human-Written Counter-Strategy asks experienced human players to write a concrete exploit-oriented action plan based on the opponent's strategy description, aiming for clarity and executability. Other baselines are implemented in our sequential-game setting following their core mechanisms, using DeepSeek-V3.2 as the base model. Refer to implementation details in Appendix B.2.

### 4.2. Main Results

We evaluate online test-time learning where the agent plays sequential games against an opponent, updating memory after each game. Table 1 summarizes performance on multi-round RPS and LHE. The results highlight three key observations.

**MEMOPILOT Delivers Consistent Gains Over Memory-**

*Table 1.* Main results on strategic games. We report mean@64 across evaluation runs. For Memory w/ MEMOPILOT, (+) denotes absolute improvement over Memory w/ Qwen2.5-14B.

| Method | Qwen2.5-14B-Instruct as Player | | Qwen3-235B-A22B as Player | |
|---|---|---|---|---|
| | RPS@5 | LHE@5 | RPS@5 | LHE@5 |
| *Baseline Methods* | | | | |
| No Memory | 0.43 | -1.36 | 0.44 | -1.46 |
| Full History | 0.02 | -1.22 | 0.03 | -1.45 |
| Human-Written Counter-Strategy | **1.0** | **1.08** | **0.57** | **0.39** |
| *Previous Methods* | | | | |
| Reflexion (Shinn et al., 2024) | 0.61 | -1.27 | 0.53 | -0.85 |
| ExpeL (Zhao et al., 2024) | 0.03 | -0.39 | -0.02 | -0.58 |
| MemoryBank (Zhong et al., 2023) | 0.43 | -0.96 | 0.48 | -1.29 |
| AWM (Wang et al., 2025c) | 0.64 | -1.17 | 0.67 | -1.32 |
| ReasoningBank (Ouyang et al., 2026) | 0.81 | -1.14 | 0.81 | -0.87 |
| *Memory-based Methods* | | | | |
| Memory w/ Qwen2.5-7B-Instruct | 0.36 | -0.13 | 0.19 | -0.85 |
| Memory w/ DeepSeek-V3.2 | 1.64 | -0.78 | 1.46 | -0.60 |
| Memory w/ Gemini-3.0-Flash | 0.45 | -1.26 | 0.54 | 1.16 |
| Memory w/ Qwen2.5-14B-Instruct | 0.21 | -0.23 | 0.34 | -0.29 |
| Memory w/ MEMOPILOT | **3.28** (+3.10) | **2.03** (+2.30) | **3.27** (+2.90) | **1.31** (+1.60) |

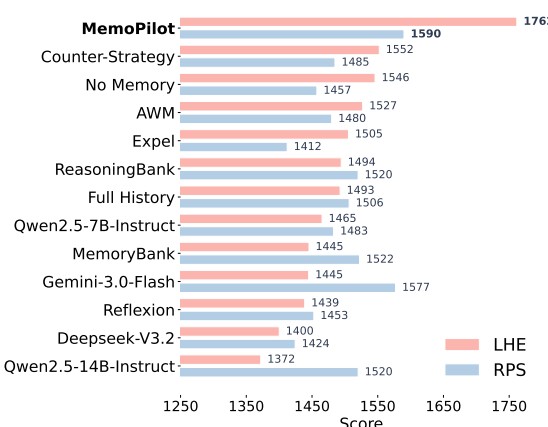

*Figure 5.* Elo ranking of memory methods on RPS and LHE computed from head-to-head matches. Higher is better.

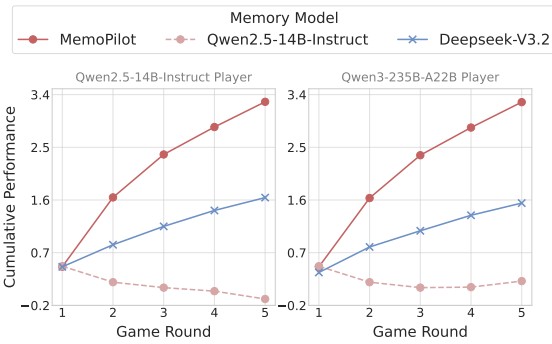

*Figure 6.* Test-time learning dynamics in RPS of MEMOPILOT compared to baseline memory models across sequential games, evaluated with two frozen players. Left: the player used during memory training (Qwen2.5-14B-Instruct). Right: zero-shot generalization to a stronger frozen player (Qwen3-235B-A22B). MEMOPILOT yields consistently higher cumulative performance and improves rapidly within a few games.

**Free and Prompting Baselines.** Across both games, MEM-OPILOT achieves the strongest average performance at five rounds. With Qwen2.5-14B as the frozen player, MEMOPI-LOT reaches 3.28 on RPS@5 and 2.03 on LHE@5, while No Memory remains at 0.43 and −1.36. Prompting-based memory baselines provide only limited improvements on RPS and are generally negative on LHE, indicating that heuristic memory updates do not reliably translate into better decisions in this online setting.

**Elo Rankings Confirm a Consistent Strength Advantage.** Figure 5 aggregates head-to-head outcomes into an Elo score for each method in both games. MEMOPILOT

ranks first on both RPS and LHE with scores of 1590 and 1762, respectively, showing a consistent advantage over prompting-based baselines and memory-free play beyond the specific @5 metric in Table 1. Implementation details are provided in Appendix B.3.

**Naively Longer Histories Can Hurt, Suggesting the Need for Selective Memory.** Full History performs poorly on RPS and stays negative on LHE in Table 1. This degradation suggests that simply appending more interaction rounds can introduce noise and dilute the actionable signal needed for the next move. In contrast, MEMOPILOT compresses experi-

*Table 2.* StreamBench results. We report overall accuracy (pass@4) averaged across all turns.

| Method | CoSQL | DS-1000 |
|---|---|---|
| No Memory | 69.5 | 50.0 |
| Full History | 70.0 | 52.5 |
| Memory w/ DeepSeek-V3.2 | 67.5 | 50.0 |
| Memory w/ Qwen2.5-14B | 66.0 | 48.8 |
| Memory w/ MEMOPILOT | **73.5** | **56.3** |

ence into a compact memory that preserves the information most relevant for future rounds.

**MEMOPILOT Improves Rapidly and Generalizes Across Frozen Players.** Beyond final-round scores, Figure 1 and Figure 6 show that MEMOPILOT improves sharply within the first few games and then continues to accumulate higher performance. The same pattern holds across different frozen players. Notably, although we train the memory model with Qwen2.5-14B-Instruct as the frozen player, it successfully assists a substantially stronger player, Qwen3-235B-A22B, achieving 3.27 on RPS@5 and 1.31 on LHE@5. These results suggest that MEMOPILOT learns a robust memory update behavior that extracts transferable strategic signals from early experience, rather than relying on brittle, model-specific prompting recipes.

### 4.3. Real-World Evaluation on StreamBench

To evaluate whether the learned memory update mechanism transfers beyond games, we extend MEMOPILOT to StreamBench (Wu et al., 2024), a benchmark for continuous improvement of language agents. We use Qwen2.5-14B-Instruct as the execution agent. The evaluation contains 32 held-out episodes, each with 5 sequential tasks sampled from the same CoSQL database or DS-1000 Python library. At each turn, the agent receives a new task, executes it, and incorporates environment feedback. We report overall accuracy (pass@4) averaged across all turns. Table 2 shows that full History provides only marginal gains over No Memory, and prompt-based memory updates with DeepSeek-V3.2 or Qwen2.5-14B-Instruct do not improve performance, while MEMOPILOT achieves the best performance on both tasks.

## 5. Analysis

### 5.1. Learned Memories Act as Executable Guidance

We study why learned memories outperform hand-crafted alternatives by isolating the form of information provided to the frozen player. Table 3 highlights a key gap between *semantic correctness* and *behavioral usefulness*. When given the ground-truth opponent strategy description, the player improves over No Memory, increasing RPS@5 from 0.43 to 0.75 and LHE@5 from −1.36 to −0.48. However, the

*Table 3.* Performance with different provided memory variants. For the rewrite condition, we use DeepSeek-V3.2 to post-edit MEMOPILOT's generated memories into more natural, professional English while strictly preserving all logic, numbers, and strategy.

| Memory Input | RPS@5 | LHE@5 |
|---|---|---|
| No Memory | 0.43 | -1.36 |
| Ground-Truth Opponent Strategy | 0.75 | -0.48 |
| Human-Written Counter-Strategy | 1.00 | 1.08 |
| MEMOPILOT | **3.28** | **2.07** |
| +Rewrite w/ DeepSeek-V3.2 | 3.12 | 1.65 |

*Table 4.* Memory format ablation on LHE@5. All trainable memory variants use the same multi-turn GRPO recipe unless marked w/o RL.

| Method | LHE@5 |
|---|---|
| No Memory | -1.36 |
| Full History | -1.22 |
| 3-tier memory w/o RL | -0.23 |
| Free-form memory w/ RL | 1.04 |
| 3-tier memory w/ RL | **2.03** |

player still struggles to translate correct facts into reliable decisions. While supplying a human-written counter-strategy helps, it remains notably weaker than MEMOPILOT: it reaches 1.00 on RPS and 1.08 on LHE, whereas MEMOPILOT achieves 3.28 and 2.07 under the same setting. This gap is consistent with the role of test-time learning: MEMOPILOT continually updates memory from game outcomes, refining hypotheses and action rules as evidence accumulates, which enables rapid adaptation over a few games and robustness to situation-dependent variations in play.

To further separate the impact of *content* from *surface phrasing*, we additionally rewrite MEMOPILOT's generated memories with DeepSeek-V3.2 into more natural, professional English while strictly preserving all logic, numbers, and strategy. This rewrite retains most of MEMOPILOT's gains (3.12 on RPS and 1.65 on LHE), and still substantially outperforms ground-truth alternatives, suggesting that the primary benefit comes from learning decision-relevant strategic content that better identifies opponent tendencies and provides effective action guidance. The remaining gap to MEMOPILOT indicates that the original phrasing and structure can further help the frozen player execute the advice. More implementation details are provided in Appendix E.1.

### 5.2. Memory Format Ablation

We isolate the effect of the structured memory format by training a free-form scratchpad variant with the same multi-turn GRPO recipe. Table 4 shows that RL training is essential, as the 3-tier memory without RL only slightly improves over Full History. Under the same RL recipe, the free-form variant improves substantially, but the 3-tier format per-

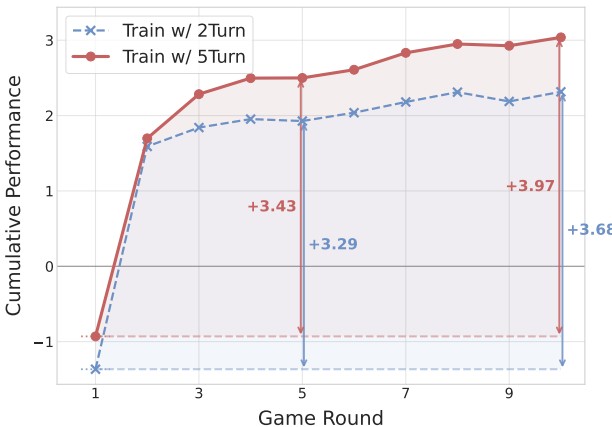

*Figure 7.* Effect of training horizon on test-time learning performance. We compare memory models trained with shorter (2-turn) vs. longer (5-turn) rollouts, and plot cumulative performance over game rounds. Longer-horizon training yields more stable gains that accumulate over extended gameplay.

*Table 5.* Cross-opponent evaluation with a single evolving memory state throughout the full interaction stream. We report performance on the last 5 games (games 6–10). For cold-start, the agent plays only 5 games against the target opponent B. For warm-up, the agent plays 5 games against opponent A (or B) before evaluating on B.

| Setting | RPS@5 | LHE@5 |
|---|---|---|
| No Memory (cold-start on B) | 0.43 | -1.36 |
| *Memory w/* MEMOPILOT | | |
| Cold-start on B (5 games only) | 3.28 | 2.03 |
| Warm-up on A, then B | 2.56 | 3.26 |
| Warm-up on B, then B | 5.22 | 3.58 |

*Table 6.* Reward design comparison: cumulative reward is the sum of rewards over future memory-guided games, while one-step reward assigns each memory update credit based only on the next following game outcome.

| Method | LHE@5 |
|---|---|
| No Memory | -1.36 |
| Cumulative Reward | 0.61 |
| One-step Reward | **2.03** |

forms better, suggesting that the structure provides a useful inductive bias for maintaining hypotheses and translating them into executable guidance.

### 5.3. Multi-Turn Training Enables Long-Horizon Stability

We also study how the *training horizon* (episode length during multi-turn GRPO training) influences long-horizon behavior. Figure 7 compares MEMOPILOT trained with 2-turn versus 5-turn rollouts for 10 consecutive games. Longer-horizon training yields consistently higher cumulative performance across game rounds, with the gap emerging after the initial exploration games and widening as evidence accumulates. This suggests that training with longer rollouts better teaches the memory model to preserve hypotheses, revise them when contradicted, and provide guidance that remains effective over extended interaction.

### 5.4. Cross-Opponent Evaluation

We further evaluate robustness under different opponents by switching opponents mid-stream. Concretely, the player model first plays 5 consecutive games against opponent A and then plays 5 games against a different opponent B, while keeping a single evolving memory state throughout. We report metrics only on the last 5 games (games 6–10). Across test opponents, we pair each evaluation opponent B with a distinct warm-up opponent A via a fixed bijection to balance coverage. Table 5 shows that MEMOPILOT remains effective after the opponent switch, achieving 3.26 on LHE, indicating that the learned memory updates can revise prior beliefs rather than overfitting to early experience.

### 5.5. Reward Design

We ablate reward design for training the memory model. We compare optimizing a cumulative return over memory-guided games with using a one-step per-turn assignment that credits each memory update $m_t$ by the next-game reward $r_{t+1}$. We find that per-turn one-step rewards are substantially more stable, while cumulative return often collapses after a certain number of training steps. Table 6 shows that the performance of cumulative reward is far below one-step reward, only 0.61 on LHE. We attribute this to variance from future environmental randomness, which pushes the memory model toward generic memories rather than context-sensitive adaptation.

### 5.6. Failure Mode Analysis

MEMOPILOT's main failure mode is a maintenance–refinement tradeoff. The hypothesize-and-verify cycle accumulates evidence to avoid overreacting to noisy individual games, but this conservatism can make memory stale when an opponent deliberately reverses behavior after the agent has committed to a counter-strategy. Table 7 evaluates such non-stationary and adaptive settings. Performance decreases when opponents switch more frequently or when the opponent is also equipped with memory, but MEMOPILOT remains substantially above the No Memory baseline.

## 6. Related Work

**Memory-Augmented Language Agents.** Equipping LLMs with memory has emerged as a key direction for building

*Table 7.* Failure-mode analysis on LHE@5 under non-stationary or adaptive opponents.

| Setting | LHE@5 |
|---|---|
| No Memory | -1.36 |
| Same opponent | 2.03 |
| Opponent switches every 5 games | 1.76 |
| Opponent switches every 2 games | 1.21 |
| Opponent with Memory (DeepSeek-V3.2) | 1.25 |

adaptive agents. Generative Agents (Park et al., 2023) introduced memory streams for social simulation. Subsequent work has explored various memory architectures: Agent Workflow Memory (Wang et al., 2025c) extracts reusable workflows from trajectories; A-MEM (Xu et al., 2025) proposes agentic memory with self-organization; MEM1 (Zhou et al., 2025) learns to synergize memory and reasoning; MemGen (Zhang et al., 2025) generates latent memory for self-evolving agents; and Buffer of Thoughts (Yang et al., 2024b) maintains thought templates for reasoning. Unlike most prior work that focuses on within-task persistence, we study *cross-game* strategic memory that must evolve across sequential matches, and we train the memory *evolve* process end-to-end to optimize downstream utility.

**Experience-Driven and Lifelong Learning.** Recent work has explored how agents can learn from accumulated experience. Reflexion (Shinn et al., 2024) uses verbal self-reflection for improvement, while ExpeL (Zhao et al., 2024) accumulates insights across tasks. Dynamic Cheatsheet (Suzgun et al., 2025) maintains evolving memory through heuristic updates; ReasoningBank (Ouyang et al., 2026) scales memory through trajectory comparison. Skill-Weaver (Zheng et al., 2025a) and PolySkill (Yu et al., 2026b) study reusable skills for self-improving or continual agents. These works are complementary to our setting: they largely rely on heuristic or prompt-based experience updates, whereas MEMOPILOT optimizes the memory update policy directly with downstream reward. Benchmarks including EvaLearn (Dou et al., 2025), LifelongAgentBench (Zheng et al., 2025b), and work measuring test-time learning with human comparison (Wang et al., 2025a) have begun systematically evaluating these capabilities. However, existing models still suffer from limitations in their ability to leverage experience for self-improvement (Huang et al., 2024; Dou et al., 2025; Suzgun et al., 2025). Our approach addresses this by providing RL training that optimizes memory quality through task performance.

**RL for Optimizing Text and Auxiliary Policies.** Reinforcement learning has been used to optimize a variety of text artifacts around LLMs. RLPrompt (Deng et al., 2022) and OPRO (Yang et al., 2024a) optimize prompts for downstream tasks. Prompt-R1 (Liu et al., 2025a) train prompt rewriters via RL. RLAD (Qu et al., 2025) trains abstrac-

tion generators for reasoning, demonstrating that allocating test-time compute to abstraction generation can outperform generating more solutions directly. Xie et al. (2025) train critics via RL using a decoupled architecture, while Advisor Models (Asawa et al., 2025) learn lightweight policies to steer black-box LLMs. SPIRAL (Liu et al., 2026) improves strategic reasoning through self-play RL on the player itself. In contrast, MEMOPILOT keeps the player frozen and trains an external memory module, making it applicable to stronger or closed-source players without player-side parameter updates. Our work follows this general paradigm of training auxiliary models with RL, but focuses specifically on strategic memory generation with multi-turn training.

## 7. Limitations

**1) Dependence on informative experience and rewards.** MEMOPILOT is designed for settings where past interactions contain reusable signal and downstream reward is available for training. When trajectories are low-information or rewards are extremely sparse, memory updates may have limited evidence to improve from. Auxiliary signals such as token efficiency or trajectory-quality rubrics could provide denser training feedback in such settings.

**2) Bounded memory capacity.** Our experiments use a 512-token memory budget for fair comparison. This budget is a hyperparameter that can be scaled with task requirements, but very long single-task trajectories may still require standard preprocessing such as chunking or summarization before memory updates.

**3) Degrade when faced with non-stationary or adaptive opponents.** As discussed in Table 7, MEMOPILOT can degrade when the environment changes faster than its evidence accumulation cycle, especially if a new opponent directly exploits a previously stored belief. This reflects a tradeoff between maintaining stable beliefs under stochasticity and rapidly refining them under distribution shifts.

## 8. Conclusion

We introduce MEMOPILOT, a framework that treats memory updating as a trainable decision process optimized via multi-turn GRPO. By using turn-level advantage estimation and proxy rewards, our approach stabilizes learning in stochastic environments and significantly outperforms heuristic baselines. On both LHE and RPS, MEMOPILOT enables frozen LLMs to achieve rapid test-time learning, ranking first in Elo ratings. Furthermore, the learned memory policies demonstrate strong robustness, successfully generalizing to unseen opponents and larger player models without additional parameter updates.

## Acknowledgments

This Work was done during the first author's internship at Zhipu AI.

## Impact Statement

This paper presents work whose goal is to advance the field of Machine Learning. There are many potential societal consequences of our work, none of which we feel must be specifically highlighted here.

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

# A. Training Algorithm

---

**Algorithm 1** Multi-Turn MEMOPILOT Training

---

**Require:** Generator $G_\theta$, player $\pi$, strategies $\mathcal{S}$, number of games per episode $T$, group size $G$
1: **for** each training iteration **do**
2:     Sample opponent strategy $\sigma \sim \mathcal{S}$
3:     **for** $i = 1$ to $G$ **do**
4:         $m_{i,0} \leftarrow \emptyset$
5:         **for** $t = 1$ to $T$ **do**
6:             $(e_{i,t}, r_{i,t}) \leftarrow \mathrm{Play}(\pi, \sigma, m_{i,t-1})$
7:             **if** $t < T$ **then**
8:                 $m_{i,t} \sim G_\theta(\cdot \mid e_{i,t}, m_{i,t-1})$
9:             **end if**
10:         **end for**
11:         **for** $u = 1$ to $T - 1$ **do**
12:             $R_{i,u} \leftarrow r_{i,u+1}$
13:         **end for**
14:     **end for**
15:     Compute $\hat{A}_{i,t,k}$ via Eq. 4
16:     Update $\theta$ by maximizing $\mathcal{J}(\theta)$ in Eq. 6
17: **end for**

---

# B. Evaluation Details

## B.1. Computational Cost

MEMOPILOT incurs a one-time training cost and a lightweight inference-time memory-update cost. During training, the dominant cost is environment rollout with the frozen player and opponent. We report the RL training hyperparameters in Table 8.

At inference time, MEMOPILOT adds one memory-update generation between consecutive interactions, i.e., $T - 1$ memory-update LLM calls over $T$ interactions. This has the same $O(T)$ update complexity as memory-based baselines such as Reflexion, ExpeL, and ReasoningBank, which also require at least one reflection or memory-update call after an interaction. MEMOPILOT does not introduce an additional critic model or dense retrieval module, and the trained memory model can be reused with unseen player models without player-side retraining.

*Table 8.* RL hyperparameters.

| Parameter | Value |
|---|---|
| Training Batch Size | 32 |
| Mini-Batch Size | 32 |
| Group Size | 16 |
| Learning Rate | $1 \times 10^{-6}$ |
| KL Coefficient | 0.0001 |
| Maximum Prompt Length | 2048 |
| Maximum Response Length | 6144 |

## B.2. Baseline Implementations in Our Setting

We implement all baselines in the same sequential-game setting as MEMOPILOT, where the agent plays multiple consecutive games against a fixed opponent and updates its cross-game memory after each game under the same memory budget. For **Full History**, we concatenate the full interaction histories from previous games as context. **Human-Written Counter-Strategy** asks experienced human players to write a concrete exploit-oriented action plan based on the opponent's strategy description, aiming for clarity and executability. For **Reflexion** (Shinn et al., 2024), after each game, we generate a short

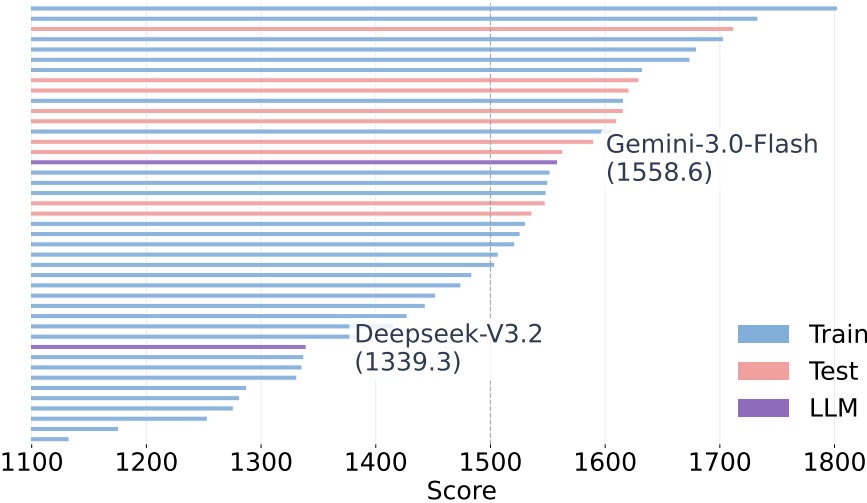

*Figure 8.* Elo ratings of LHE opponent strategies estimated from head-to-head matches, illustrating that our constructed opponent pool spans a broad and relatively uniform difficulty range. Blue and pink bars denote training and held-out opponents, respectively, while purple bars denote LLMs, while purple bars denote LLMs.

reflection and append it as memory for the next game. For **ExpeL** (Zhao et al., 2024), after each game, we extract a concise experience statement (success/failure insight) and accumulate these as memory. For **MemoryBank** (Zhong et al., 2023), we store past game summaries and retrieve relevant items to form the memory input. For **AWM** (Wang et al., 2025c) and **ReasoningBank** (Ouyang et al., 2026), we follow their core mechanisms to maintain and update reusable patterns across games. All previous method baselines use DeepSeek-V3.2 as the base model.

### B.3. Elo-Based Difficulty Calibration

We use standard Elo updates. Each method is assigned a rating $R$ initialized to 1500. For a head-to-head matchup between method $i$ and an opponent $j$, the expected score is

$$E_i = \frac{1}{1 + 10^{(R_j - R_i)/400}}, \tag{7}$$

and the rating update takes the form

$$R_i \leftarrow R_i + K\,(S_i - E_i), \tag{8}$$

where $K$ is a fixed step size and $S_i \in [0, 1]$ is an outcome score derived from the empirical head-to-head results.

For the difficulty calibration figures (e.g., Figure 8), we estimate Elo for opponent strategies via round-robin matches. For the method rankings in Figure 5, we evaluate each memory-based method against all held-out test opponents. Each method–opponent pair is played for $k = 5$ consecutive games with memory updates enabled; to focus on test-time learning rather than the initial no-memory exploration, we compute $S_i$ using only the memory-enabled games (games 2–5).

## C. Opponent Strategy Construction and Verification

A key design choice in our framework is the construction of a diverse and controllable opponent pool that enables systematic study of test-time learning capabilities. We collect the opponent pool to satisfy three core principles:

**Controllability via Executable Instructions.** Rather than relying on black-box opponent models, we equip each opponent with an executable instruction. This design choice provides two critical advantages: (i) *reproducibility*, allowing us to generate consistent multi-game trajectories for stable RL training, and (ii) *interpretability*, enabling us to verify that the memory model learns to identify and exploit genuine strategic patterns.

**Behavioral Diversity through Systematic Variation.** We construct strategy *families* that systematically vary along interpretable behavioral axes. For LHE, these axes include action-frequency biases (calling stations vs. folders), phase-specific aggression patterns (e.g., turn/river-focused pressure), and deceptive modes (e.g., check-raise traps). For RPS,

we design strategies spanning open-loop sequences, one-step reactive rules conditioned on recent history, and multi-step counter-patterns.

**Mechanism-Based Train-Test Separation.** We separate train and test sets by the underlying *mechanism*. Held-out opponents preserve strategic intent while shifting surface realizations, decision triggers, or the phases where information is revealed. For example, in LHE, test opponents over-represent street-specialized strategies (e.g., passive early but sudden turn pressure) and trigger-based adaptations (delayed steals, river bluffs), which stress-test whether memory can localize *when* a strategy deviates and update guidance accordingly. In RPS, held-out strategies emphasize rule compositions with conditional triggers and edge cases (e.g., multi-trigger policies, parity-based rules), requiring hypothesis maintenance and revision as evidence accumulates.

**Construction Pipeline.** We construct a controllable pool of opponent strategies to systematically evaluate test-time learning and to ensure that different opponents correspond to meaningfully different behaviors. First, we recruit experienced human players to write a small set of seed strategies in natural language, covering representative playing styles and common exploitable patterns. Second, we use LLM-based rewriting to (i) standardize strategies into a consistent instruction format, (ii) expand each seed into multiple variants with different hyperparameters or conditional branches, and (iii) add edge-case clarifications to improve executability. Third, we manually verify and iterate: annotators review each strategy text for coherence and implementability, run short pilot games under the intended prompting configuration, and check whether observed actions match the specified policy. When deviations are detected (e.g., inconsistent action frequencies or violations of hard constraints), we either revise the strategy description and re-test, or drop the strategy if it remains unstable. This pipeline yields 32 training RPS strategies, 45 training LHE strategies, and 41 held-out strategies (32 RPS, 9 LHE) for generalization evaluation.

## C.1. Strategy Cases

### RPS Case 1 (reactive counter rule).

```
ABSOLUTE COMMAND: You are Player 0. Obey this strategy strictly.

History parsing: In the history, 'Opponent (P0) played X' is your own past move. 'You (P1) played Y' is the opponent
    's past move.

Strategy (reactive):
- Round 1: play [scissors].
- Round 2-6: if the opponent's last move BEAT your last move, then play the counter to the opponent's last move.
    Otherwise, repeat your last move.
```

### RPS Case 2 (deterministic lookup table).

```
ABSOLUTE COMMAND: You are Player 0.

Strategy: Use a deterministic lookup table keyed by (your last move, opponent last move).
- Round 1: [rock].
- Round 2-6: Let A be your last move and B be opponent last move. Play:
  If (A,B) is (rock,rock)->[paper], (rock,paper)->[rock], (rock,scissors)->[scissors];
  If (paper,rock)->[scissors], (paper,paper)->[rock], (paper,scissors)->[paper];
  If (scissors,rock)->[rock], (scissors,paper)->[paper], (scissors,scissors)->[scissors].
```

### LHE Case 1 (Turn Aggressor).

```
You are a professional Poker player. You are playing 2-player Limit Texas Hold'em. Your player identity is specified
    by the environment/system message. Strategy: Turn Aggressor. Preflop and Flop: Play very passively, 'check'
    and 'call' only. Turn: If you have any Pair or a Draw, 'raise' aggressively to the cap. Many players fold on
    the Turn when facing unexpected aggression. Always choose from legal_actions and output only {'action': ''}.
```

### LHE Case 2 (River Bluff Specialist).

```
You are a professional Poker player. You are playing 2-player Limit Texas Hold'em. Your player identity is specified
    by the environment/system message. Strategy: River Bluff Specialist. Play passively (check/call) on Flop and
    Turn. On the River, if the board shows any possible Straight or Flush and you have NOTHING, 'raise' to the cap.
    This represents a huge finished hand. Always choose from legal_actions and output only {'action': ''}.
```

# D. Prompting Details

### D.1. Prompt Templates

**3-tier adaptive memory (cross-game).** The memory model is prompted as a *Game Strategy Curator* that maintains an evolving 3-tier memory across repeated games. The prompt enforces a structured "Hypothesis–Verify–Confirm" update loop and separates (i) **REASONING** (analysis of evidence and outcome), (ii) **KNOWLEDGE_MAINTENANCE** (a compact hypothesis state with counters and evidence tags), and (iii) **FINAL_STRATEGY_PROMPT** (actionable rules for the next game). Only the `<final_strategy_prompt>` content is visible to the player, while the full `<cheatsheet>` is carried over to the next turn. The complete prompt template is shown below.

**Prompt Template for 3-Tier Adaptive Memory**

```
# GAME STRATEGY CURATOR: 3-TIER ADAPTIVE MEMORY

### Purpose and Goals
You are a sophisticated Game Strategy Curator. Your goal is to manage a 3-tier
    ↪ memory system that evolves through an iterative 'Hypothesis-Verify-Confirm'
    ↪ loop to exploit Player 0's behavioral patterns.

### The 3-Tier Memory Architecture
Your output must maintain these three distinct sections inside the <cheatsheet>
    ↪ block:

1. **REASONING (The 'Think' Layer)**:
    - Analyze Player 0's strategy based on the current log.
    - Compare current actions with previous hypotheses.
    - **Key Task**: Deduce why the previous round's performance was good or bad. Was
        ↪ it the strategy or just luck?

2. **KNOWLEDGE MAINTENANCE (The 'State' Layer)**:
    - Maintain a list of strategic hypotheses and their current statuses: [Hypothesis
        ↪  | Verified | Confirmed].
    - Use **Observation Counters**: Track how many times a pattern has been seen.
    - **Evidence Tagging**: Reference specific rounds (e.g., [R1], [R2]) to preserve
        ↪ long-term information without raw log storage.

3. **FINAL STRATEGY PROMPT (The 'Action' Layer)**:
    - This is the ONLY section provided to the player (Player 1) for the next round.
    - Distill everything into clear, actionable, and concise rules.

### Memory Update Format
Use the following structure:
<cheatsheet>
<reasoning>
  [Your deep analysis of Player 0's strategy and the last round's outcome]
</reasoning>
<knowledge_base>
  <item id="1">
    <pattern>[Observed behavior]</pattern>
    <status>[Hypothesis/Verified/Confirmed]</status>
    <evidence>[Ref: R1, R2. Observation Count: X. Success Count: Y]</evidence>
  </item>
</knowledge_base>
<final_strategy_prompt>
  [The distilled, actionable instructions for Player 1]
</final_strategy_prompt>
</cheatsheet>

N.B. Only the <final_strategy_prompt> content is seen by Player 1, but the ENTIRE <
    ↪ cheatsheet> is passed to you in the next round. Content not explicitly copied
    ↪  from the previous cheatsheet will be lost. You MUST explicitly preserve and
    ↪ update all relevant tiers.
```

# E. Additional Details

## E.1. Memory Input

*Ground-Truth Opponent Strategy* provides the frozen player directly with the opponent's strategy used to instantiate that opponent in our evaluation.

We also include a post-processing baseline that rewrites MEMOPILOT's generated memories into more natural, professional English while keeping the strategic content unchanged. Concretely, after MEMOPILOT produces a memory, we send the memory text to DeepSeek-V3.2 with the following instruction and then provide the rewritten memory (and only the rewritten memory) to the frozen player. All other evaluation settings are kept identical.

```
Please rewrite the following AI-generated strategic memory into natural, professional English. You must keep all
    logic, numbers, and strategy identical, but remove any robotic or repetitive phrasing to make it read like a
    human expert's advice. Do not add or remove any information.
```

# F. Qualitative Examples

## F.1. Multi-turn Memory Evolution Example (MEMOPILOT)

---

**Turn 1 (MEMOPILOT)**
**Playthrough**

```
Hand 1: In this Limit Texas poker game, there are 2 players from 0 to 1. Player 0's hand is ['CT', 'H4'].
    Player 1's hand is ['D2', 'H3']. The community card is ['S9', 'C9', 'CK', 'C2', 'D5']. Preflop. Player
    1 calls. Player 0 checks. Flop. Player 1 checks. Player 0 checks. Turn. Player 1 checks. Player 0
    raises. Player 1 folds. Player 0 wins by 1.0 chip(s). Player 1 loses by 1.0 chip(s).

Hand 2: In this Limit Texas poker game, there are 2 players from 0 to 1. Player 0's hand is ['D2', 'H3'].
    Player 1's hand is ['CT', 'H4']. The community card is ['S9', 'C9', 'CK', 'C2', 'D5']. Preflop. Player
    0 raises. Player 1 calls. Flop. Player 0 checks. Player 1 checks. Turn. Player 0 raises. Player 1 calls
    . River. Player 0 raises. Player 1 calls. Player 0 wins by 6.0 chip(s). Player 1 loses by 6.0 chip(s).

Reward: -7.0
```

**Memory**
```
None
```

---

**Turn 2 (MEMOPILOT)**
**Playthrough**

```
Hand 1: In this Limit Texas poker game, there are 2 players from 0 to 1. Player 0's hand is ['S2', 'SA'].
    Player 1's hand is ['SQ', 'CA']. The community card is ['SJ', 'D7', 'C5', 'D3', 'DJ']. Preflop. Player
    1 raises. Player 0 calls. Flop. Player 1 raises. Player 0 calls. Turn. Player 1 raises. Player 0 raises
    . Player 1 raises. Player 0 raises. Player 1 calls. River. Player 0 raises. Player 1 raises. Player 0
    raises. Player 1 raises. Player 0 calls. Player 0 loses by 19.0 chip(s). Player 1 wins by 19.0 chip(s).

Hand 2: In this Limit Texas poker game, there are 2 players from 0 to 1. Player 0's hand is ['SQ', 'CA'].
    Player 1's hand is ['S2', 'SA']. The community card is ['SJ', 'D7', 'C5', 'D3', 'DJ']. Preflop. Player
    0 raises. Player 1 raises. Player 0 calls. Flop. Player 1 raises. Player 0 raises. Player 1 raises.
    Player 0 calls. Turn. Player 1 raises. Player 0 raises. Player 1 raises. Player 0 raises. Player 1
    calls. River. Player 0 raises. Player 1 raises. Player 0 raises. Player 1 raises. Player 0 calls.
    Player 0 wins by 22.0 chip(s). Player 1 loses by 22.0 chip(s).

Reward: -3.0
```

**Memory**
```
<cheatsheet>
    <reasoning>
        The last strategy prompt was partially effective. In Hand 1, Player 1 did not fold to the opponent's
            raises and was more aggressive, which led to a win. However, in Hand 2, despite being more
            aggressive, Player 1 did not win, indicating that the opponent's aggression might have been
            more effective in that instance. The overall pattern of the opponent's behavior (raising too
            much on Flop and Turn) was correctly identified.
    </reasoning>
    <knowledge_base>
        <entry>
        The opponent raises too much on the Turn and Flop, often after checks.
```

```
        <status>Verified</status>
        <evidence>Hand 1: Flop -> Turn -> Opponent raises and forces a fold.
        </evidence>
        </entry>
    <entry>
        The opponent is overly aggressive with raises, especially when Player 1 checks.
    <status>Confirmed</status>
    <evidence>
    Hand 2: Turn -> River -> Opponent raises after Player 1 calls.
    </evidence>
    </entry>
    </knowledge_base>
    <final_strategy_prompt>
    - Do not fold to opponent's raises.
    - Raise more often on Flop and Turn.
    - Do not let opponent dictate the pace; be more aggressive.
    - Be prepared to re-raise after opponent's raises. </final_strategy_prompt>
</cheatsheet>
```

## Turn 3 (MEMOPILOT)
### Playthrough

```
Hand 1: In this Limit Texas poker game, there are 2 players from 0 to 1. Player 0's hand is ['H4', 'SQ'].
    Player 1's hand is ['S5', 'H6']. The community card is ['DA', 'C6', 'CT', 'H2', 'SK']. Preflop. Player
    1 raises. Player 0 calls. Flop. Player 1 raises. Player 0 raises. Player 0 raises. Player 1 raises.
    Player 0 raises. Player 1 calls. Turn. Player 0 raises. Player 0 raises. Player 1 raises. Player 0
    raises. Player 1 raises. Player 0 raises. Player 0 raises. Player 1 raises. Player 0 raises. Player 1
    raises. Player 0 loses by 19.0 chip(s). Player 1 wins by 19.0 chip(s).

Hand 2: In this Limit Texas poker game, there are 2 players from 0 to 1. Player 0's hand is ['S5', 'H6'].
    Player 1's hand is ['H4', 'SQ']. The community card is ['DA', 'C6', 'CT', 'H2', 'SK']. Preflop. Player
    0 raises. Player 1 ls. Flop. Player 1 raises. Player 0 raises. Player 1 raises. Player 0 raises. Player
     1 calls. Turn. Player 0 raises. Player 1 raises. Player 0 raises. Player 1 raises. Player 0 folds.
    Player 0 loses by 13.0 chip(s). Player 1 wins by 13.0 chip(s).

Reward: 32.0
```

### Memory
```
<cheatsheet>
        <reasoning>
        The last strategy prompt was effective. Both In Hand 1 and Hand 2, Player 1 did not fold to the
            opponent's raises and was more aggressive, which led to a win. The overall pattern of the
            opponent's behavior (raising too much on Flop and Turn) was correctly identified.
        </reasoning>
        <knowledge_base>
        <entry>
        The opponent raises too much on the Turn and Flop, often after checks.
        <status>Confirmed</status>
        <evidence>
        Hand 1: Flop -> Turn -> Opponent raises and forces a fold.
        </evidence>
        </entry>
        <entry>
        The opponent is overly aggressive with raises, especially when Player 1 checks.
        <status>Confirmed</status>
        <evidence>
        Hand 2: Turn -> River -> Opponent raises after Player 1 checks.
        </evidence>
        </entry>
        </knowledge_base>
        <final_strategy_prompt>
        - Do not fold to opponent's raises.
        - Raise more often on Flop and Turn.
          - Do not let opponent dictate the pace; be more aggressive.
        - Re-raise after opponent's raises.
        - Do not fold preflop; be aggressive.
          - Be prepared to call or raise after opponent's aggressive actions.
        </final_strategy_prompt>
</cheatsheet>
```

