# OpenReview forum: "From Player to Master: Enhancing Test-Time Learning of LLM Agents via Reinforcement Learning over Memory"
_ICML.cc/2026/Conference — ICML 2026 regular_

### Official Review · Reviewer_BA4q · 2026-03-02

**Soundness:** 3
**Presentation:** 2
**Significance:** 2
**Originality:** 2
**Overall Recommendation:** 4
**Confidence:** 2

**Summary:**

The authors address a question of whether explicit, trainable memory policies can unlock reliable across‑game improvement for LLM agents, grounding their work on test-time learning (TTL). During the test-time learning (TTL) agent improves its actions by using the accumulated past experience. To enable the TTL it is suggested to use the dynamic memory updates. Currently most of the approaches use handmade or prompt-based update rules and there is insufficient research that considers TTL as a trainable parameter. The authors propose  MEMOPILOT --  a framework based on training the memory update process within consecutive agentic interactions.  They convert memory update process into a multy-turn decision problem, and further optimize it using GRPO.  Multy-turn training simulates the “hypothesize-and-verify” approach. Evaluation of MEMOPILOT on Rock–Paper–Scissors (RPS) and Limit Texas Hold’em (LHE) demonstrates consistent increase of rewards during  inference-time learning. MEMOPILOT substantially outperforms “no memory”, “full history”, human counter strategies, and several memory based baselines (Reflexion, ExpeL, MemoryBank, AWM, ReasoningBank) in both per game metrics and Elo rankings, and the learned memory model transfers to a stronger frozen player without retraining. In overall, the paper presents a technically sound and well‑conducted study of RL‑trained memory updates for TTL in LLM agents, with strong improvements over a range of baselines. The main drawbacks are the limited domain scope (only RPS and LHE) and somewhat minimal discussion of limitations, and lack of real-world applications scenarios.

**Compliance With Llm Reviewing Policy:**

Affirmed.

**Key Questions For Authors:**

1.	I wonder if MEMOPILOT can be used in real-world TTL scenarios (e.g., tool‑use agents, code debugging,etc)? How experiments can be extended to explore further real-world applications, what challenges do you foresee that might need to adjust the MEMOPILOT (e.g. unclear rewards in real-world scenarios, longer horizons, etc.)?
2.	Did you compare the current three‑tier memory format (diagnosis / maintenance / guidance) with simple alternatives (e.g., single “notes”, or guidance only) under the same RL training?
3.	Could you report the number of RL steps/episodes, total games simulated per environment, and approximate training cost (e.g., GPU‑hours, LLM calls) for MEMOPILOT, and compare this to the cost of running baseline methods?
4.	You show that using the next game’s outcome as the reward for each memory update is more stable than using the full cumulative return over future games (Table 4). What about intermediate designs, such as giving each memory update credit for the next few games (e.g., 2–3) instead of just the next-game or cumulative reward? I am just wondering if the performance drops gradually with the increase of the horizon, or maybe it can peak at 3-rd, 4-th, etc. game? It might not be the case. since, as it is mentioned in the text, due to the stochasticity of the learning signal for long-horizon returns, but still it will be interesting to see.
5.	Many baselines use DeepSeek‑V3.2 as the underlying LLM, while MEMOPILOT uses Qwen2.5‑14B as teh base model. Have you cosnidered experiments where Reflexion/ExpeL/ReasoningBank are also instantiated with Qwen2.5‑14B instead of DeepSeek‑V3.2 to observe the method vs. base‑model effects? What about other LLMs families like Llama or Mistral?
6.	MEMOPILOT uses Qwen2.5‑14B as the base model.  Have you explored how MEMOPILOT behaves with different backbone LLMs (e.g., DeepSeek V3.2 itself, smaller Qwen variants, or other families). I would like to see how model agnostic is the proposed approach.

**Limitations:**

Yes

**Strengths And Weaknesses:**

Soundness.

Strengths:

The methodology is well formulated and presented in a relatively acceptable way: the multi-turn memory generation and memory updating process is well-formulated, the authors clearly explain and justify representation of the memory space as three‑tier structure. The 3 principles-based agent design process, following by the human‑in‑the‑loop pipeline with LLM-based standardization and manual verification is well explained and justified. Authors ensure equal difficulty for test and train pools by estimating Elo ratings for each opponent strategy. The experiments including MEMOPILOT evaluation on two strategic games and  comparison against a set of baselines are well designed. Results and analysis section are well presented and explained.

Weaknsees:

1.	On Figure 2 the authors mention the multi-agent arena, however, the paper discusses only two‑player games. It is unclear if the fundings can be extended to the multi-agent environments.

2.	While RPS and LHE provide controllable frameworks, they are still small and limited environments.

3.	The paper is based on the “frozen player assumption”. However, what if the player is changing?I mean further fine‑tuned or RL‑trained during deployment. Similarly, the assumption is the player is stateless across games. What if the player has a state (e.g. player that keeps its own long‑term memories independent of MEMOPILOT’s memory)?

4.	The overall framework is correct for one‑step reward design, and it is justified by the instability of credit assignment and increased environment noise due to stochastic learning signal for long-horizon returns.  But I wonder if authors considered how to deal with these problems and explored whether the memory- updating process can be  adapted to predict the reward for the next steps?

5.	The computational complexity of the proposed framework is not discussed: the number of training iterations, wall‑clock time, and total games simulated per environment aren’t provided.

6.	There is no ablation of the three‑tier memory structure (diagnosis / maintenance / guidance) vs. simpler formats; it is unclear whether performance gains are coming mostly form RL training, from the hand‑crafted structure, or their interaction.

7.	It is unclear why particularly Deepseek V3.2 is used to interpret the memories generated by MEMOPILOT. A short justification will not hurt. Also, how authors ensure that other model, say, Mistral, will provide the same interpretations as Deepseek?

8.	Qwen2.5 14B Instruct as the base model of MEMOPILOT, however there is no justification on why this particular family of LLMs is selected as baseline. How the results will change if some other family is used as the based model? In other words, I want to see how model‑agnostic is the MEMOPILOT.

Presentation.

Strengths:

The paper is well structured and written in a relatively acceptable, though hard to read in some parts, language. Most of the assumptions and steps are well-justified. The TTL motivation to MEMOPILOT design and the experimental validation is coherent.
Figures and tables are informative: Figure 1 and Figure 6 show learning curves over rounds for different memory models and 2 different frozen-LLMs based players. Figure 2  gives a general overview of the proposed framework. Figure 3 clearly explains memory-updating process. Figure 5 compares  relative strengths across methods on two games (LHE and RPS).
Appendices contain additional details on Elo computation, baseline instantiations, and opponent strategy examples, along with the prompting template and RL hyper parameters.

Weaknesses:

1.	While theoretical part of the paper is coherent, some mathematical notation and indices require careful re-reading (n.b. despite selecting confidence level 2, math/other details were carefully checked). A small example showing  several episodes with states, actions, and rewards would make the training logic more clear. Maybe paper will benefit more of a nice visual of several episodes (instead or combined with Figure 3)?

2.	The main text gives a brief description of the 3‑tier memory format and appendix contains more practical details. Given teh conceptual importance of 3‑tier memory representation, a schematic, similar to Figure 3, in the main text would be very useful (e.g., short example of memroy evolution across two games).

3.	Poor arrangement of figures: Figure 1 is mentioned first on p. 7., Figure 2 and Figure 3 are not mentioned in the text at all, Figure 6 is on p.6  but mentioned first on p. 7.

4.	Abstract contains unexplained abbreviation for Group Relative Policy Optimization (GRPO).

5.	The paper will greatly benefit from discussing possible applications of MEMOPILOT and future work.

Significance.

Overall, I would characterize the significance as solid, but not transformative.

Strengths:

The leveraging past experience to improve the performance of the LLM agents is an important research direction. The current memory updating methods that use hand-designed or prompt-based memory update rules leading to failure of agent to consistently improve during multiple interactions. Authors address this gap by training teh memory update process, leading to substantial empirical performance e.g. against Qwen2.5-14B-Instruct baseline inproving  RPS@5 from 0.21 to 3.28 and LHE@5 from −0.23 to 2.03. It suggest that during the training of the memory update the agent learns non-trivial, model‑agnostic abstractions of strategy, leading to improved overall performance.  The construction of the opponents and selecting train-test splits based on the Elo rankings are themselves a useful contribution for evaluating TTL in controlled environments.

Weaknsees:

1.	The experiments are limited to two games and use a “frozen‑player + memory copilot” architecture.  I wonder how these designs can be transltaed to more realistic agent systems, such as web agents, tool-using agents, code assistants, etc.)

2.	Lack of discussion of possible applications of MEMOPILOT in real-world scenarios. After reading the paper I am not persuaded that I really need the TTL-based memory updates in real-life applications, given its apparent high computational cost.


Originality.

Strengths:

The authors approach to consider memory updating process as a trainable RL policy optimizing it as multi‑turn GRPO is quite a novel application of known methods. In contrast to the prior works which used human-crafted heuristic or prompt‑based memory updates rather than RL‑trained memory policies.

Weaknesses:

1.	The methodologies and tools used in the paper are well known, therefore the paper does not propose a new algorithmic framework and can be cosnidered as a novel application of known technologies.

2.	The three‑tier memory structure is not compared with teh existing memory frameworks, therefore it is hard to tell how original is this approach.

3.	The method relies on a fixed size, text only memory that must fit in teh LLM’s context window of 512 tokens, which may limit its applicability to longer horizon agents where the context cannot be easily compressed into a few hundred tokens.

---

> ### Author Rebuttal · Authors · 2026-03-31
>
> Dear Reviewer BA4q, we are genuinely grateful for your valuable feedback. We are encouraged by your positive comments on our **well-structured writing**, **well-formulated methodology**, **novel application**, **well-designed experiments**, and **important research direction**. We hope our clarifications below can address your concerns.
>
> ---
>
> **W1**: Figure 2 mentions multi-agent, but only discusses two-player games.
>
> **R1**: We will revise Figure 2. We also evaluated on **3-player LHE** (LHE@5): No Memory: -1.84, Full History: -1.13, MemoPilot: **0.68**. MemoPilot maintains a substantial advantage despite increased uncertainty.
>
> ---
>
> **W2**: RPS and LHE are limited environments.
>
> **R2**: We extended to **StreamBench**, a complex real-world benchmark. **MemoPilot achieves the best performance on both tasks (+4.0 on CoSQL, +6.3 on DS-1000 over No Memory)**. See R1 to reviewer 8ZcC for full results.
>
> ---
>
> **W3**: What if the player changes during deployment or has its own long-term memory?
>
> **R3**: MemoPilot is not limited to the frozen-player setting. We conducted the following experiments:
>
> - **Player-shift**: the player switches from Qwen2.5-14B to Qwen3-235B at turn 6. LHE@5 on the last 5 games: No Memory (235B only) -1.42, MemoPilot (235B only) 1.56, MemoPilot (Player-shift 14B→235B) **1.98**. Accumulated memory provides a warm-start advantage even across model switches.
> - **Stateful player**: we layered MemoPilot on top of ExpeL (LHE@5): ExpeL alone -0.39, MemoPilot alone **2.03**, ExpeL + MemoPilot 1.59. MemoPilot remains dominant even atop another memory system.
>
> ---
>
> **W4**: What about intermediate reward horizons (e.g., 2–3 steps)?
>
> **R4**: We tested intermediate designs with a 5-turn rollout:
>
> | Reward Horizon | LHE@5 |
> | --- | --- |
> | 1-step | **2.50** |
> | 2-step | 1.39 |
> | 3-step | 1.35 |
> | 4-step | 1.20 |
>
> Performance degrades monotonically, confirming that longer horizons amplify noise and destabilize credit assignment. Predicting future rewards is inherently difficult due to unpredictable game states. Two promising directions: **(a)** rubric-based memory quality evaluation (dense, state-independent signal), and **(b)** learned value function conditioned on memory state. We will discuss in our revision.
>
> ---
>
> **W5**: Could you report RL steps, total games simulated, and training cost, and compare this to the cost of running baseline methods?
>
> **R5**: MemoPilot trains for 200 RL steps, totaling 100k games per environment (~2.05M LLM calls, ~8 GPU-hours on 8 H100s). This moderate one-time investment yields substantial gains, and the trained module transfers zero-shot to unseen players (R3). At inference, MemoPilot uses only $T-1$ LLM calls over $T$ turns, which shares the same $\mathcal{O}(T)$ complexity as all memory-based baselines (Reflexion, ExpeL, ReasoningBank).
>
> ---
>
> **W6**: No ablation of memory format.
>
> **R6**:  Under the same GRPO recipe: No Memory -1.36, Full History -1.22, 3-tier w/o RL -0.23, free-form w/ RL 1.04, **3-tier w/ RL 2.03**. RL training is essential (w/o RL: -0.23), and **the 3-tier structure provides effective inductive bias over free-form (2.03 vs 1.04)**. We will add comparison with existing memory frameworks in the revision.
>
> ---
>
> **W7**: Why DeepSeek for interpreting memories and as base model of baselines, and Qwen2.5-14B as base model of MemoPilot? How model-agnostic is MemoPilot?
>
> **R7**:
>
> - DeepSeek-V3.2 was used as a strong reference for baselines. Results are robust to interpreter choice: Gemini-3.0-Flash achieves 1.90 LHE@5 vs. 1.65 with DeepSeek-V3.2.
> - Fair baseline comparison with Qwen2.5-14B: Reflexion -0.97, ExpeL -1.47, MemoPilot **2.03**. The gap remains large regardless of model choice.
> - Qwen2.5-14B was chosen for wide adoption and suitable RL training size. We also trained with Llama-3.1-8B-Instruct, achieving 1.32 LHE@5 (vs. 0.17 without RL), confirming RL-trained memory benefits generalize across model families.
>
> ---
>
> **W8**: Presentation issues. The 512-token memory may limit applicability to longer-horizon agents.
>
> **R8**: We will fix presentation issues. 512 tokens suffice for current domains; the memory length is a **hyperparameter** that can be scaled up.
>
> ---
>
> **W9**: How does MemoPilot translate to realistic agent systems? What challenges are foreseen?
>
> **R9**: **MemoPilot is broadly applicable to scenarios requiring efficient TTL**. By reducing trial-and-error costs, it effectively tackles private-domain tasks (e.g., internal codebases) and personalized adaptation where pre-training falls short. Our StreamBench results validate this. Cost is low since inference only adds T-1 LLM calls (R5).
>
> Challenges: **(1) Sparse rewards**: auxiliary signals (e.g., token efficiency) could supplement binary success. **(2) Low-information trajectories**: when past interactions carry little signal (e.g., trivial tasks), memory updates have limited room for improvement; detecting and skipping uninformative turns could help. We will expand this discussion.

---

> > ### Author Rebuttal · Reviewer_BA4q · 2026-04-03
> >
> > Thank you for the detailed rebuttal, most of my concerns were addressed. I would suggest including the full StreamBench setup/results in the revision, including tasks, baselines, whether MEMOPILOT is trained on or transferred to StreamBench, and the exact evaluation protocol.

---

> > > ### Author Response · Authors · 2026-04-04
> > >
> > > Dear Reviewer BA4q:
> > >
> > > Thank you so much for the positive feedback and the constructive suggestion! We will make sure to include the complete StreamBench experimental details in the revised version.
> > >
> > > Once again, thank you very much for your dedication and for the effort in reviewing our paper!

---

### Official Review · Reviewer_Cc3g · 2026-03-11

**Soundness:** 3
**Presentation:** 3
**Significance:** 3
**Originality:** 2
**Overall Recommendation:** 5
**Confidence:** 4

**Summary:**

This paper proposes MemoPilot, a plug-in memory copilot module that enables LLMs to perform test-time learning by explicitly maintaining and updating a structured memory across game turns. The key contribution is formulating memory-augmented multi-turn interaction as an MDP and extending GRPO to the multi-turn setting with turn-wise reward signals and turn-level advantage estimation. The memory system has three tiers: Identification (detect opponent type), Maintenance (track behavioral patterns), and Guidance (derive counter-strategies). MemoPilot is evaluated on Rock-Paper-Scissors (6-round matches) and Limit Texas Hold'em (LHE), achieving top Elo ratings (1590 RPS, 1762 LHE) and outperforming baselines including approaches using ground-truth opponent strategy information.

**Compliance With Llm Reviewing Policy:**

Affirmed.

**Final Justification:**

We thank the authors for the thorough analysis of failure modes and the illustrative examples. The maintenance-refinement tradeoff framing and the concrete seed-level comparison directly address our concern. We are satisfied with the response and raise our score accordingly. Please incorporate the missing citations, limitations, and failure modes in the final version.

**Key Questions For Authors:**

1. Does MemoPilot provide any benefit over a simple free-form scratchpad (no tier structure) trained with the same multi-turn GRPO procedure? This would isolate the contribution of the three-tier design from the general benefit of having writable memory.

2. Recent work such as SPIRAL [3] shows that directly applying multi-turn RL with self-play on zero-sum games can internalize strategic reasoning without an explicit memory module. How does MemoPilot compare against simply RL-training the model on the game setting directly, without the external memory copilot? What is the added value of the memory architecture over letting the model internalize opponent adaptation through RL alone?

3. What is the inference-time cost of the memory copilot compared to other memory-based methods? Does the extra LLM call per turn for memory updates justify the performance gains over cheaper alternatives?

[3] Liu, B., Guertler, L., Yu, S., Liu, Z., Qi, P., Balcells, D., ..., & Jaques, N. "SPIRAL: Self-Play on Zero-Sum Games Incentivizes Reasoning via Multi-Agent Multi-Turn Reinforcement Learning." arXiv:2506.24119, 2025. https://arxiv.org/abs/2506.24119

**Limitations:**

The paper does not include a limitation part, which could be discussed whether the three-tier memory structure introduces brittleness when facing opponent types not seen during training — the cross-opponent results (Table 3) are encouraging but limited to opponents generated by the same construction pipeline. The inference-time cost of the memory copilot (an extra LLM call per turn) is never quantified, making it difficult to assess practical deployment tradeoffs. Additionally, the paper does not explore failure modes: when does MemoPilot's memory become stale or misleading (e.g., against opponents that deliberately shift strategy mid-game to exploit the memory), and how gracefully does performance degrade?

**Strengths And Weaknesses:**

### Strengths

1. **Interesting finding**: The result that learned memory outperforms ground-truth opponent strategy descriptions (Table 2) is the paper's strongest contribution. It suggests the model learns task-relevant compressed representations rather than merely storing complete information. This is genuinely insightful.

2. **Opponent construction pipeline**: The human-in-the-loop approach with controllability, behavioral diversity verification, and mechanism-based train-test separation is a thoughtful experimental design that addresses a real challenge in multi-agent evaluation.

3. **Reward design analysis**: Table 4 comparing one-step vs. cumulative rewards provides useful empirical guidance for multi-turn RL, showing that one-step reward significantly outperforms cumulative reward due to credit assignment difficulties.

4. Solid and good writing: The paper covers most memory baselines and shows improvement upon, and it's also written in a clear manner, which is easy to read.

### Weaknesses

1. **Evaluation domains are trivially simple**: Rock-Paper-Scissors — even the 6-round variant — is an extremely simple game with 3 actions and memoryless Nash equilibrium. Limit Hold'em is better but still a solved game with known optimal strategies. It is unclear whether MemoPilot's benefits extend to genuinely complex domains. The paper also does not compare against simpler memory alternatives (e.g., a free-form scratchpad, appending interaction history to context), so we cannot attribute the gains to the specific three-tier architecture vs. the general benefit of any memory mechanism.

2. **Missing related work on lifelong learning agents**: The paper does not cite several closely related works. SkillWeaver [1] demonstrates agents that self-improve by discovering and honing reusable skills — both are directly relevant to the memory-driven adaptation paradigm MemoPilot targets; and [2] propose polymorphic skill abstraction for generalizable lifelong learning. These omissions leave a gap in the paper's positioning relative to existing work on adaptive, self-improving agents.

3. The proposed method might incur higher costs compared to other baselines. Reporting the cost of each method, including memory updates, would help justify the method’s benefits.

[1] Zheng, B., Fatemi, M. Y., Jin, X., Wang, Z. Z., Gandhi, A., Song, Y., ..., & Su, Y. "SkillWeaver: Web Agents can Self-Improve by Discovering and Honing Skills." arXiv:2504.07079, 2025. https://arxiv.org/abs/2504.07079

[2] Yu, S., Li, G., Shi, W., & Qi, P. "PolySkill: Learning Generalizable Skills Through Polymorphic Abstraction For Continual Learning" ICLR 2026. https://openreview.net/forum?id=KdEsujyiSV

---

> ### Author Rebuttal · Authors · 2026-03-31
>
> Dear Reviewer Cc3g, we sincerely thank you for your valuable time and comments. We are encouraged by your positive comments on our **interesting finding**, **thoughtful experimental design**, **reward design analysis**, and **solid and good writing**. We hope our clarifications below address your concerns.
>
> ---
>
> **W1**: Evaluation domains are trivially simple. It is unclear whether MemoPilot's benefits extend to genuinely complex domains. The paper also does not compare against simpler memory alternatives.
>
> **R1**: For broader-domain results, see R1 to 8ZcC (**StreamBench**: CoSQL text-to-SQL +4.0, DS-1000 coding +6.3 over No Memory, both best among all methods). We ablate **memory formats** under the same GRPO recipe:
>
> | Method | LHE@5 |
> | --- | --- |
> | No Memory | -1.36 |
> | Full History | -1.22 |
> | MemoPilot w/o RL training | -0.23 |
> | MemoPilot w/ free-form | 1.04 |
> | MemoPilot w/ 3-tier | **2.03** |
>
> Key findings: (1) **RL training is essential**—without RL, the 3-tier structure yields only marginal gain over Full History. (2) **The 3-tier structure provides useful guidance**, constraining the search space to meaningful memory updates. Under the same recipe, 3-tier (2.03) substantially outperforms free-form (1.04).
>
> ---
>
> **W2**: Missing related work on lifelong learning agents.
>
> **R2**: We will add SkillWeaver and PolySkill to the related work. Both maintain reusable experiences across interactions; the key distinction is how knowledge is updated: they rely on heuristic or prompt-based rules, while MemoPilot optimizes end-to-end via RL with downstream task reward. As shown in our results, RL-trained updates substantially outperform prompt-based ones under different memory structures. We view the two paradigms as complementary and will clarify the positioning in the revision.
>
> ---
>
> **W3**: The proposed method might incur higher costs compared to other baselines. What is the inference-time cost of the memory copilot compared to other memory-based methods? Does the extra LLM call per turn for memory updates justify the performance gains over cheaper alternatives?
>
> **R3:** MemoPilot uses $T-1$ LLM calls for memory updates over $T$ turns, which is the minimal requirement and shares the same $\mathcal{O}(T)$ complexity as baselines (Reflexion, ExpeL, ReasoningBank) that also require at least one LLM call per game. Because MemoPilot is designed without complex additional modules like dense retrieval, its actual overhead is comparable or even lower. A detailed empirical cost comparison will be added to the revision.
>
> ---
>
> **W4**: Recent work such as SPIRAL [3] shows that directly applying multi-turn RL with self-play on zero-sum games can internalize strategic reasoning without an explicit memory module. How does MemoPilot compare against simply RL-training the model on the game setting directly, without the external memory copilot? What is the added value of the memory architecture over letting the model internalize opponent adaptation through RL alone?
>
> **R4**: The key distinction is in **what is being optimized**. SPIRAL and similar approaches use RL to improve the player's task-specific competence, whereas MemoPilot trains a **test-time learning (TTL) module** that enables a player to learn from experience at inference time. We focus not on how to make a model better at playing games, but on how to make a model better at *learning to play games* from interactions in cross-game settings. The game setting is a controlled testbed but not limited to it.
>
> This distinction has concrete practical implications:
>
> | | SPIRAL (Player RL) | MemoPilot (Memory RL) |
> | --- | --- | --- |
> | Trainable component | Player weights | Memory module only |
> | Transfer to new player | Requires retraining | Zero-shot plug-in (even for strong player like Qwen3-235B) |
> | Closed-source player | Not applicable | Fully supported |
>
> ---
>
> **W5**: The paper does not include a limitation part, which could be discussed whether the three-tier memory structure introduces brittleness when facing opponent types not seen during training — the cross-opponent results (Table 3) are encouraging but limited to opponents generated by the same construction pipeline. Additionally, the paper does not explore failure modes: when does MemoPilot's memory become stale or misleading (e.g., against opponents that deliberately shift strategy mid-game to exploit the memory), and how gracefully does performance degrade?
>
> **R5**: We conducted experiments under different settings:
>
> | Setting | LHE@5 |
> | --- | --- |
> | Same opponent | 2.03 |
> | Opponent switch every 5 games | 1.76 |
> | Opponent switch every 2 games | 1.21 |
> | Opponent with Memory (DeepSeek-V3.2) | 1.25 |
>
> MemoPilot performs well under rapid switching and memory-equipped opponents, still significantly outperforming No Memory.
>
> We will add a detailed case study and limitation analysis in the revision.

---

> > ### Author Rebuttal · Reviewer_Cc3g · 2026-04-01
> >
> > Thank you for all the additional experiments and clarifications, and most of my issues are resolved.
> >
> > However, I would still like to see a limitations section and a discussion of failure modes. Specifically: when does MemoPilot's memory become stale or misleading? For example, against opponents that deliberately shift strategy mid-game to exploit accumulated memory (e.g., establishing a predictable pattern early, then reversing it once the agent has committed to a counter-strategy). No additional experiments are needed — I am interested in the authors' analysis of how the memory mechanism handles such adversarial non-stationarity, and under what conditions performance degradation is expected.
> >
> > Once these discussions are included by the authors, I am happy to improve my score accordingly.

---

> > > ### Author Response · Authors · 2026-04-02
> > >
> > > Dear Reviewer Cc3g, thank you very much for the valuable feedback. We sincerely hope our clarifications below can address your concerns.
> > >
> > > **The central limitation is that MemoPilot's evidence accumulation mechanism trades responsiveness for robustness.** The "hypothesize-and-verify" cycle accumulates evidence across games to filter noise, making it effective under stochastic but learnable environments. However, when an opponent deliberately reverses behavior, this same conservatism delays adjustment: MemoPilot does not overwrite a belief after a single contradictory game, because individual games can be noisy or unrepresentative. Under stationary play this avoids overreacting to variance; under deliberate strategy reversal, stale beliefs persist until enough contradictory evidence accumulates. We refer to this as the **maintenance-refinement tradeoff**.
> > >
> > > Performance degradation is most pronounced when this tradeoff is stressed in the following ways:
> > >
> > > **(1) When the opponent switches to a strategy that directly exploits the stored belief.** Not all switches cause failure — degradation is most severe when the new strategy directly punishes the committed counter-strategy. If the confirmed belief remains roughly effective against the new opponent, performance can be partially preserved.
> > >
> > > **(2) When the switch happens faster than the evidence accumulation cycle.** MemoPilot needs multiple interactions to distinguish a true strategy shift from normal stochasticity, as each game provides only partial and noisy evidence. If the opponent changes before enough contradictory evidence is observed, stale guidance persists.
> > >
> > > **(3) When the belief has already been promoted to confirmed status before the shift.** A hypothesis verified through consistent early evidence is harder to overturn than one still being tested. Once promoted, subsequent contradictory observations must accumulate before adjustment is triggered.
> > >
> > > **Concrete example.** We examine two episodes facing the identical opponent sequence: Turn-Check-Raise Trapper (passive early streets, establishing a predictable pattern) → Showdown Seeker (never folds, calls everything — reversing the pattern once the agent has committed). Different seeds yield different dealt hands, leading to different early evidence and belief states at the time of the shift.
> > >
> > > *Failure case (seed 21) — belief already verified before the shift:*
> > >
> > > | Turn | Opponent | Reward (chip) | Memory (excerpt) |
> > > | --- | --- | --- | --- |
> > > | 2 | Trapper (passive) | +12.0 | "passive players **fold to aggressive moves**; pattern verified" |
> > > | 3 | Showdown Seeker | -6.0 | "continue aggressive strategy to **exploit passive folding tendencies**" |
> > >
> > > Consistent early evidence promotes the hypothesis to verified status. After the shift, the Showdown Seeker calls every raise, but the verified belief persists — the mechanism treats the contradictory outcome cautiously, as it could reflect an unrepresentative hand rather than a genuine strategy change.
> > >
> > > *Recovery case (seed 2, same transition) — belief still in hypothesis stage:*
> > >
> > > | Turn | Opponent | Reward (chip) | Memory (excerpt) |
> > > | --- | --- | --- | --- |
> > > | 2 | Trapper (passive) | +3.5 | "aggressive-passive style observed...adopt aggressive style to **test the opponent's patience**" |
> > > | 3 | Showdown Seeker | -8.0 | **Adds branch**: "**If the opponent responds aggressively**, be prepared to capitalize on their aggressive style" |
> > >
> > > Here, the belief remains in the hypothesis-testing stage ("test the opponent's patience"). When contradictory evidence arrives, it is naturally incorporated as part of ongoing testing — a new branch is added rather than requiring a confirmed belief to be overturned.
> > >
> > > The contrast illustrates condition (3): the speed of adjustment after a strategy shift depends on where in the "hypothesize-and-verify" cycle the current belief sits. Beliefs still under testing incorporate contradictory evidence naturally; beliefs already promoted to confirmed status require more accumulated evidence before adjustment.
> > >
> > > In the final version, we will add a dedicated **Limitations and Failure Modes** discussion covering this analysis.
> > >
> > > Thank you again for your valuable time and constructive comments!

---

### Official Review · Reviewer_8ZcC · 2026-03-13

**Soundness:** 2
**Presentation:** 2
**Significance:** 1
**Originality:** 2
**Overall Recommendation:** 3
**Confidence:** 5

**Summary:**

This paper proposes MemoPilot, a framework that optimizes LLM's multi-turn game playing with memory update and multi-turn GRPO.

**Compliance With Llm Reviewing Policy:**

Affirmed.

**Key Questions For Authors:**

See Weakness 1

**Limitations:**

See Weakness 1

**Strengths And Weaknesses:**

Strength:

1. The method use principled multi-turn GRPO framework, and conducts thorough Elo evaluation, showing significant performance gain on the LHE and RPS.

Weaknesses:

1. The RPS and LHE games are too simple. Especially for RPS, by just choosing actions randomly without any strategy, both players could achieve an optimal 1/3 win, draw and lose rate. The flawed evaluation significantly limits the method's practical utility. I recommend adding multi-turn dialogue, or other more complex games to further demonstrate multi-turn memory advantages.

---

> ### Author Rebuttal · Authors · 2026-03-31
>
> Dear Reviewer 8ZcC, we sincerely thank you for your precious time and valuable comments. We are encouraged by your positive comments on our **principled framework**, **thorough evaluation**, and **significant performance gains**. We sincerely hope our clarifications and new experiments below can address your concerns.
>
> ---
>
> **W1**: The RPS and LHE games are too simple. Especially for RPS, by just choosing actions randomly without any strategy, both players could achieve an optimal 1/3 win, draw and lose rate. The flawed evaluation significantly limits the method's practical utility. I recommend adding multi-turn dialogue, or other more complex games to further demonstrate multi-turn memory advantages.
>
> **R1**: We would like to clarify why RPS, despite its apparent simplicity, serves as a meaningful testbed for TTL. The Nash equilibrium argument applies to self-play between rational agents. Our evaluation measures a fundamentally different capability: **exploiting sub-optimal opponents via test-time adaptation**, which is the core of TTL. A random policy cannot exploit the diverse reactive strategies in our opponent pool — only memory-driven adaptation can. Two strong pieces of evidence support this: **(1)** MemoPilot **outperforms providing the player with ground-truth opponent strategy** (Table 2). This is impossible to explain if the domain were trivially simple, and demonstrates that the learned memory captures non-trivial, opponent-specific abstractions beyond what explicit strategy descriptions convey. **(2)** Elo-based difficulty calibration shows that the opponent pool covers a wide difficulty range; even SOTA LLMs like Gemini-3.0-Flash rank low, confirming the pool is far from trivial.
>
> Following your suggestion, we extended MemoPilot to **StreamBench [1]**, a complex real-world benchmark for continuous improvement of language agents, including `CoSQL` (text-to-SQL) and `DS-1000` (Python data-science coding). In our experimental setup, we deployed Qwen2.5-14B-Instruct as the execution agent. The evaluation consists of 32 held-out episodes, each containing 5 sequential tasks randomly sampled from the same CoSQL database or DS-1000 Python library. At each turn, the agent receives a novel task, executes the task and incorporates environment feedback. Training follows the default configuration detailed in our paper. Performance is measured by the overall accuracy (pass@4) averaged across all turns. The results are summarized below:
>
> | Task | No Memory | Full History | Memory w/ DeepSeek-V3.2 | Memory w/ Qwen2.5-14B-Instruct | Memory w/ MemoPilot |
> | --- | --- | --- | --- | --- | --- |
> | CoSQL | 69.5 | 70.0 | 67.5 | 66.0 | **73.5** |
> | DS-1000 | 50.0 | 52.5 | 50.0 | 48.8 | **56.3** |
>
> Two key observations: **(1)** Full History barely improves over No Memory (e.g., 70.0 vs 69.5 on CoSQL), confirming that raw history alone is insufficient — selective, learned memory is necessary. **(2)** Prompt-based memory (DeepSeek-V3.2, Qwen2.5-14B-Instruct) does not consistently outperform Full History, while **MemoPilot achieves the best performance on both tasks**, validating that RL-trained memory generalizes beyond game settings to complex real-world tasks.
>
> References: [1] Wu et al. "StreamBench: Towards Benchmarking Continuous Improvement of Language Agents." NeurIPS 2024.

---

### Official Review · Reviewer_6MDH · 2026-04-03

**Soundness:** 3
**Presentation:** 3
**Significance:** 3
**Originality:** 2
**Overall Recommendation:** 4
**Confidence:** 4

**Summary:**

This paper propose MEMOPILOT to addresses the limitation of memory systems in LLM agents that trains the memory update process using Reinforcement Learning. By formulating memory evolution as a multi-turn decision problem and optimizing it via a specialized multi-turn GRPO recipe, the authors enable a frozen LLM player to improve its performance through sequential interactions. The training introduces a turn-wise reward signal and context-independent advantage estimation to stabilize credit assignment. Evaluated on strategic games like Rock-Paper-Scissors , MEMOPILOT achieves SOTA Elo ratings and demonstrates impressive zero-shot generalization across different frozen player models.

**Compliance With Llm Reviewing Policy:**

Affirmed.

**Final Justification:**

My major concern is the task involved in this work. The two games seems too easy to demonstrate further generalization, extremely in tasks like real world application: Webshop, AlFWorld. However, comprehensive experiments are conducted to demonstrate the effectiveness of the proposed method from multiple perspectives, such as generalization. Therefore, I recommend it to weak acceptance.

**Key Questions For Authors:**

Is the method also effective in real world tasks like these involved in Openclaw?

**Limitations:**

Yes

**Strengths And Weaknesses:**

**Strength**
1. This work proposes a high-performing agent training framework, validated on game tasks.
2. The framework can continuously self-improve by leveraging experience, aligning memory updates more closely with downstream objectives.
3. Comprehensive experiments are conducted to demonstrate the effectiveness of the proposed method from multiple perspectives, such as generalization.
4. The writing is clear and easy to follow, with good presentation.

**Weakness**
1. The performance on real-world scenarios remains unknown, such as tasks Webshop or AlfWorld.
2. The current trajectories and memories are presumably relatively short; as the token length continues to grow, the memory model seems to struggle with handling longer contexts. For example, in some real-world scenarios, trajectories typically contain hundreds of thousands or even millions of tokens.
3. There is a lack of training cost comparison across methods. Compared to vanilla GRPO, how much additional computational overhead does this method introduce?

---

### Decision · Program_Chairs · 2026-04-30

**Decision:**

Accept (regular)

**Comment:**

This paper proposes **MemoPilot**, a framework for improving test-time learning in LLM agents by training a memory-update policy with reinforcement learning while keeping the underlying player model frozen. The central idea is to treat memory evolution across interactions as a multi-turn decision problem and optimize it end-to-end with a tailored GRPO-based training recipe. Reviewers generally agreed that the paper addresses an important and timely problem, namely how to move beyond hand-designed or prompt-based memory updates toward trainable memory mechanisms that better align with downstream objectives. The paper was viewed as technically sound, clearly written, and supported by strong empirical improvements over a broad set of baselines.

The main concern raised by multiple reviewers was the limited scope of the original evaluation, as well as other concerns on the lack of the discussion of failure modes, positioning in the literature and free-form memory ablation. The rebuttal has helped to address the concerns.

Overall, the paper makes a solid and novel contribution on trainable memory for test-time learning. I would recommend accept. In the final version, the authors should incorporate all promised revisions, especially the real-world StreamBench results, the free-form memory ablation, the training-cost discussion, the additional related work, and a clear limitations/failure-modes section.